# Lights and Shadows of Paracentesis: Is an Ultrasound Guided Approach Enough to Prevent Bleeding Complications?

Marta Patturelli [1], Luca Pignata [1], Pietro Venetucci [2] and Maria Guarino [1,*]

1   Gastroenterology and Hepatology Unit, Department of Clinical Medicine and Surgery, University of Naples Federico II, Via Sergio Pansini 5, 80131 Naples, Italy
2   Advanced Biomedical Sciences Department, University of Naples Federico II, 80131 Naples, Italy
*   Correspondence: maria.guarino86@gmail.com; Tel./Fax: +39-081-7464746

**Abstract:** Paracentesis is a validated procedure for diagnosing and managing ascites. Although paracentesis is a safe procedure with a 1–2% risk of complications such as bleeding, it is necessary to inform the patient about the possible adverse events. We would like to share our experience with two cases of bleeding after paracentesis. In our unit, two major hemorrhagic complications occurred in 162 procedures performed over the year 2020 (frequency of bleeding complications: 1.2%). We report two clinical cases of post-paracentesis abdominal wall hematomas. Despite a similar clinical presentation, the management approach was different: in the first case, embolization of the epigastric artery supplying the hematoma was performed. In the second case, conservative treatment was adopted. Our report aims to provide food for thought about a potentially challenging hemorrhagic complication, even with the risk of adverse outcomes.

**Keywords:** paracentesis; abdominal wall hematoma; refractory ascites; arterial embolization

## 1. Introduction

Large volume paracentesis is considered as a safe procedure with minimal complications risk and rarely causes morbidity or mortality. The most common complications of the procedure are ascitic fluid leakage, hemorrhage, infection, and intestinal perforation. Regarding significant bleeding, the most common etiologies are abdominal wall hematoma and hemoperitoneum.

## 2. Methods

We would like to present two cases of post-paracentesis hemorrhagic complications. We aim to describe conservative and interventional approaches to post-paracentesis abdominal wall hematoma and related differences in the clinical evolution and hospital length of stay in two patients with advanced liver disease.

## 3. Case Reports

### 3.1. First Case

An 84-year old Caucasian male patient with a past medical history of cryptogenic advanced decompensated chronic liver disease and clinically significant portal hypertension with refractory ascites (RA) treated with periodic large volume paracentesis (LVP) was admitted to our hospital to perform an abdominal paracentesis. The patient had several comorbidities: obesity, hypertensive heart disease, Jak-2-positive Philadelphia-negative chronic myeloproliferative disorder, permanent atrial fibrillation, and stage 3b chronic kidney disease. Ultrasound-guided abdominal paracentesis was performed without immediate complications with the removal of 6 L of ascitic fluid over 3 h, with the current albumin administration. Immediately after the end of the procedure, the patient started complaining of intense abdominal pain at the puncture site. The abdominal examination

revealed a palpable, painful, firm, non-pulsatile abdominal mass. No alteration in the hemodynamic parameters was found. A bedside abdominal ultrasonography showed a voluminous (7 cm) complex formation compatible with an abdominal wall hematoma. He then underwent an abdomen and pelvis CT scan angiography that showed, in the context of the muscular planes of the left iliac fossa, the presence of a coarse collection of blood density of bilobed morphology, showing active contextual bleeding in the arterial phase. Although this bleeding did not show direct continuity with an arterial vascular axis, it was strictly contiguous to the distal middle third of the inferior epigastric axis (Figure 1a–g). In light of active bleeding, we performed a percutaneous transcatheter arterial embolization. The portion of the epigastric artery supplying the hematoma was reached through selective catheterization, and a 3 × 5 mm spiral was positioned using a microcatheter. The final ultrasound check did not show hematoma replenishment. The re-evaluation with color doppler confirmed the absence of signal within the hematoma. In the following days, the patient showed stable clinical conditions with a progressive reduction of pain at the site of the hematoma and no alteration in the blood-chemical tests. He was discharged five days after the event. The following month, a transjugular intrahepatic portosystemic shunt (TIPS) placement was performed. Six months after the bleeding event, the patient died because of SARS-CoV-2 infection.

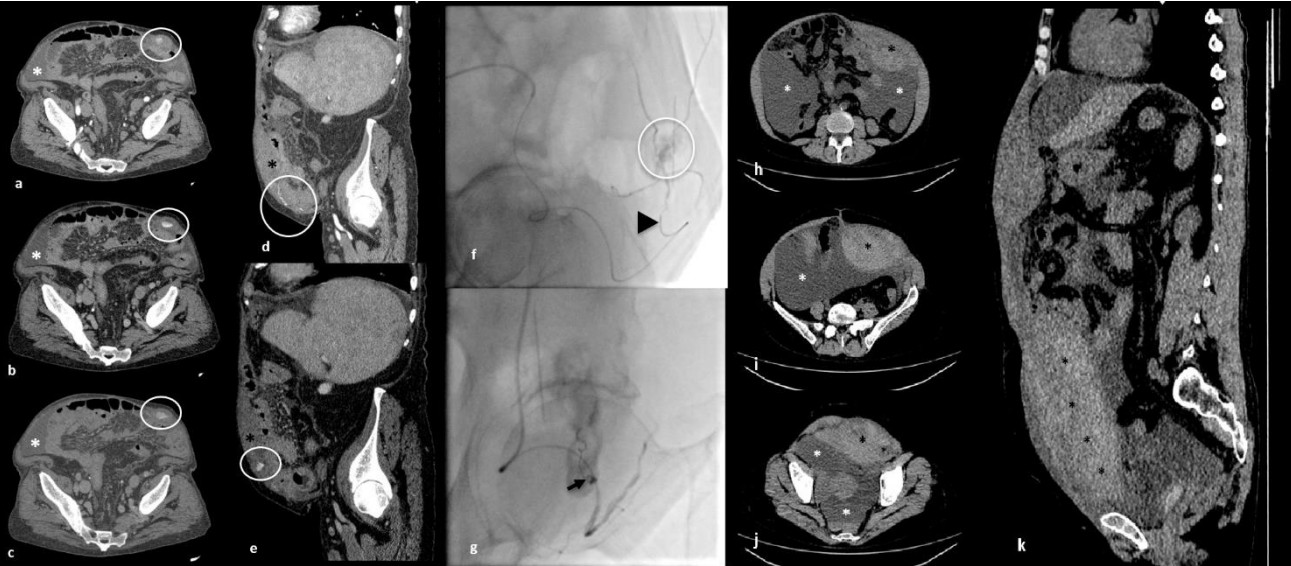

**Figure 1.** Imaging findings of the two patients. First patient's contrast-enhanced CT images: axial view of arterial (**a**), venous (**b**) and delayed (**c**) phases; sagittal CT-multiplanar reconstruction (MPR) (**d**,**e**); diagnostic angiography (**f**); and transcatheter arterial embolization (**g**). Ascites (white asterisk); extensive anterior abdominal wall hematoma (black asterisk) with evidence of active bleeding (white circle); angiographic microcatheter (black arrowhead); metallic spiral (black arrow). Second patient's non-contrast-enhanced CT images: axial scans (**h**,**i**,**j**) and sagittal MPR (**k**). Conspicuous ascitic collections (white asterisk); extensive anterior abdominal wall hematoma (black asterisk).

### 3.2. Second Case

A 51-year old Caucasian female patient with a past medical history of cryptogenic advanced decompensated chronic liver disease and RA was admitted to our hospital go undergo an abdominal LVP. Clinically significant portal hypertension was reported, with mild hypertensive portal gastropathy, splenomegaly, esophageal varices F2 with no red wale markings, and a previous hemorrhagic event requiring endoscopic ligation. The patient had no relevant comorbidities. Ultrasound-guided abdominal paracentesis was performed with the removal of 6.5 L of ascitic fluid over 4 h with the current administration of albumin. Upon the extraction of the catheter, evidence of cutaneous bleeding and progressive organization of a subcutaneous collection were found. The abdominal examination

revealed a palpable, painful, mobile, non-pulsatile abdominal mass. There was a mild drop in blood pressure and an increase in heart rate. The ultrasound bedside control showed a bulky abdominal wall hematoma, apparently replenished. After performing a compression dressing, the patient underwent an abdomen and pelvis CT scan angiography, showing the presence of an extensive blood collection in the context of the anterolateral wall of the abdomen in the subfascial area measuring about 14 × 7.5 cm in the axial section and extended in a craniocaudal direction from the subcostal seat to the pelvic excavation (Figure 1h–k). In light of the significant hemoglobin drop, the patient received two units of packed red blood cell (PRBC) transfusion and parenteral antifibrinolytic therapy. In the following days, an intravenous iron infusion, a fresh frozen plasma transfusion, parenteral antifibrinolytic therapy and further PRBCs transfusions were needed. An ultrasound-guided contralateral diagnostic paracentesis showed hemorrhagic ascitic fluid.

Multiple ultrasound bedside controls were performed, showing substantial stability of the hematoma. One week after the event, an episode of melena occurred: an emergency gastroscopy showed esophageal active variceal bleeding, which was immediately treated with endoscopic ligation. After two days, blood-chemical tests revealed a stage 1 acute kidney injury (AKI). The patient was then treated with intravenous hydration, albumin infusion, and diuretic therapy suspension, restoring normal kidney function within 48 h. She was a candidate for TIPS placement. She underwent the procedure two weeks after the abdominal wall hematoma. The day before the procedure, a therapeutic paracentesis was performed with the removal of 5 L of sero-hematic ascitic fluid. After TIPS placement, the patient developed bacterial pneumonia. The culture tests performed on biological material (blood cultures, sputum) were negative; therefore, she was treated with empiric antimicrobial therapy (meropenem and ceftobiprole). Twenty days after the event, another therapeutic paracentesis was performed with the removal of 5 L of yellow citrine ascitic fluid. She was discharged in good clinical conditions 27 days after the onset of the post-paracentesis complication. The patient initially refused the transplant option, and several months later, she agreed to undertake the evaluation process for inclusion on the transplant list. Twenty months after the bleeding event, the patient underwent liver transplant, and currently, she is in very good clinical conditions and is performing the post-transplant follow-up.

## 4. Discussion

When an invasive medical procedure is performed, the expectation is to obtain an advantage for the patient. Sometimes an adverse event occurs, defined as an unintended procedure complication that could result in harm, disability, or death for a patient. Despite the awareness that every procedure is fraught with risks, we could find ourselves unprepared to deal with the possibility of a diagnostic or therapeutic failure. We aim to discuss two post-paracentesis bleeding complications that occurred at our unit in patients with cryptogenic advanced decompensated chronic liver disease.

Abdominal paracentesis is a validated procedure for diagnosing and managing ascites of any etiology. Ascites are the most common complication of cirrhosis, and can occur in 5–10% of patients with compensated cirrhosis per year with a negative impact on quality of life and increased risk of hospitalization or further complications such as spontaneous bacterial peritonitis, renal failure, breathing or feeding discomfort, and abdominal hernias [1,2]. Adebayo reports in his review that up to 11.4% of patients with ascites will develop RA, defined as ascites that cannot be mobilized or the early recurrence of, which cannot be prevented because of loss of response to diuretic therapy or therapy-related adverse events [3].

The development of ascites, its recurrence, and the progression of underlying liver disease are all associated with poor prognosis and decreasing survival rates in the absence of liver transplant [2–4].

Diagnostic paracentesis is mandatory at the first presentation of abdominal effusion or in patients with known causes of liver disease to exclude overlapped complications such as infections or malignancies. Therapeutic paracentesis, defined as LVP when the amount of

fluid drained is at least 5 L, can be considered either beside diet and medical treatment in moderate-large ascites and recurrent ascites or even as the next step to the pharmacological approach in the case of tense ascites and refractory ascites [1]. Treatment options for RA should include LVP, insertion of TIPS, automated low-flow ascites pump (alfapump®), peritoneovenous shunt (PVS), and permanent indwelling peritoneal catheter (PIPC) [4]. The development of RA is associated with poor prognosis and decreasing survival rates in the absence of liver transplant [3,4]. LVP with albumin administration, performed as an outpatient procedure, demonstrated a higher efficacy and safety profile than diuretic therapy [5]; the latter is burdened by consequences such as renal damage, impaired hydro electrolytic balance, hemodynamic dysfunction, and precipitation of other complications of portal hypertension such as hepatic encephalopathy [4,5].

Paracentesis performed for therapeutic purposes is a safe procedure with a reported 1–2% risk of overall complications: ascitic fluid leakage from the puncture site, bleeding, infections, and bowel perforation [6,7].

Hemorrhages are among the most frequent complications of paracentesis and can occur as abdominal wall hematomas, pseudoaneurysms, and hemoperitoneum [5]. Clinically significant hemorrhage can be defined as blood loss causing a decrease in hematocrit level, associated hypotension or tachycardia, or the need for blood transfusion [8]. Rarely, post-paracentesis hemorrhages may represent life-threatening conditions [5,9,10], and even some anecdotal events of death are reported in the literature, albeit mortality in a cirrhosis setting is often conditioned by the underlying disease rather than to hemorrhage directly [10–13]. It should be emphasized that any-cause bleeding events are hazardous and hard to manage in patients with advanced chronic liver disease.

Throughout 2020, only two major hemorrhagic complications occurred in 162 procedures performed (frequency of bleeding complications: 1.2%) in our unit. Both patients had cryptogenic decompensated advanced chronic liver disease and clinically significant portal hypertension with RA treated with periodic paracentesis. The clinical features of the two patients are reported in Table 1.

Numerous contributions in the literature have investigated the factors related to increased post-procedural bleeding risk. An association between higher bleeding risks and advanced stages of cirrhosis expressed by Child–Pugh and MELD (Model for End-Stage Liver Disease) score has been studied [7,14]. De Gottardi et al., in a prospective study, evaluated the rates of overall complications on 515 paracenteses performed for cirrhosis-related ascites; major hemorrhagic complications, defined as complications requiring medical or surgical interventions, arose at 1%.

The highest probabilities of complications were observed in patients with advanced liver disease (Child–Pugh C) [15]. Lin demonstrated a high risk of hemorrhage (2.99%) in patients with acute or chronic liver disease and that low fibrinogen levels were an independent predictor of bleeding in patients with MELD > 25 [10].

An association between post-paracentesis hemorrhage and kidney impairment has also reported, possibly ascribed to platelet dysfunction [14,16]. In the setting of hospitalized patients for a post-procedure hemorrhagic event, acute kidney injury prior to paracentesis was the only independent factor in predicting risk for bleeding in decompensated cirrhotic subjects, irrespective of MELD score, amount of fluid drained, sepsis, platelets level, INR, and hemoglobin values, as reported by Hung in a retrospective analysis [17]. According to this evidence, although coagulative parameter impairment is frequent in cirrhotic patients, paracentesis is a safe procedure: the risk of bleeding complications for subjects with INR > 1.5 and PLT < 50,000/dL is around 1% [3]. However, despite a lack of correlation between the hemorrhagic risk and the platelet count or coagulopathy, it should be noted that these alterations can worsen bleeding already in progress [18].

**Table 1.** Clinical features of the two patients.

| | Patient 1 | Patient 2 |
|---|---|---|
| **Sex** | Male | Female |
| **Age** | 84 years | 51 years |
| **Liver disease** | Cryptogenic advanced chronic liver disease Child–Pugh score C10 at admission | Cryptogenic advanced chronic liver disease Child–Pugh score B9 at admission |
| **Portal hypertension-associated complications** | Moderate portal hypertensive gastropathy; splenomegaly; isolated gastric varices (IGV1); refractory ascites | Mild portal hypertensive gastropathy; splenomegaly: esophageal varices (F2 RWM-); refractory ascites |
| **Comorbidities** | Obesity; Hypertensive heart disease; Jak-2-positive Philadelphia-negative chronic myeloproliferative disorder; permanent atrial fibrillation; Stage 3b chronic kidney disease | No relevant comorbidities |
| **Ascitic fluid appearance and volume** | Sero-hematic—6 L | Yellow citrine—6.5 L |
| **Hemodynamic parameters** Before paracentesis After paracentesis | Regular No alterations found | Regular Mild drop in blood pressure and increase in heart rate |
| **Hemoglobin Value (g/dL)** Before paracentesis After paracentesis (30 min) After paracentesis (2 h) | 14.2 13.5 12.9 | 11.9 9.7 7.3 |
| **Abdominal ultrasonography findings** | A voluminous (7 × 2 cm) abdominal wall hematoma | A bulky abdominal wall hematoma, apparently replenished |
| **Abdomen and pelvis CT scan angiography findings** | A coarse collection of blood density in the context of the muscular planes of the left iliac fossa, showing contextual active bleeding strictly contiguous to the distal third of the inferior epigastric artery | A large blood collection in the context of the anterolateral wall of the abdomen in the subfascial area measuring about 14 × 7.5 cm in the axial section |
| **Therapeutic approach** | Percutaneous transcatheter arterial embolization | Conservative approach |
| **Adverse events during hospitalization** | No relevant adverse events | Anemia requiring multiple transfusions (6 units of packed red blood cells); esophageal variceal bleeding treated with endoscopic ligation and subsequent TIPS placement; episode of stage 1 acute kidney injury (AKI), treated with endovenous idratation, albumin infusion, and diuretic therapy suspension with the restoration of normal kidney function within 48 h; hospital-acquired bacterial pneumonia treated with an empirical antimicrobial therapy (after TIPS placement) |
| **Hospital length-of-stay** (days) | 5 | 27 |
| **TIPS placement** | Yes (1 month after the bleeding complication) | Yes (14 days after the bleeding complication, during hospitalization) |

Regarding the poor correlation between standard hemostatic screening and bleeding risk, it must be emphasized that the dosage of serum factors that investigate hemostasis (platelet count, INR, coagulation factors) has some limitations in cirrhotic patients. Abnormal laboratory test values in these subjects may correspond to an effective hemostatic

function. This concept, known as rebalanced hemostasis, is due to the simultaneous alteration of both the anti-thrombotic and pro-coagulant components. However, this unstable equilibrium leads cirrhotic patients to frequent hemorrhagic or thrombotic complications. Standard laboratory tests have partially explored the assessment of hemostasis in this precarious scenario. Consolidated evidence in the literature reports how this limit can be overcome with additional evaluations of the hemostatic function. Thromboelastography (TEG) and rotational thromboelastometry (ROTEM) are two viscoelastographic evaluation devices available as real-time tests at the point-of-care useful for evaluating the complete hemostatic dynamics derived from the interaction between platelets, blood cells, and plasma on whole blood: clot formation, clot strength, and clot lysis. These methods have proven to be valid in predicting the risk of bleeding in cirrhotic patients undergoing an invasive procedure, exceeding the standard hemostatic screening in accuracy [19].

Bleeding after paracentesis is due in most cases to traumatic lacerations of the abdominal wall venous or arterial vessels because of needle insertion. In cirrhotic patients, portal hypertension and variceal formation may alter the normal anatomy of collateral branches of the abdominal wall's major vessels, counteracting the role of anatomical landmarks in individuating an appropriate puncture site. Moreover, the vessels can be displaced and stretched because of the ascitic distension of the abdomen or overweight/obesity, becoming more exposed to injuries [20]. The most common cause of hemorrhage is the injury of the inferior epigastric artery (IEA) or its branches [21–23]. However, lesions of other vessels such as deep circumflex inferior artery (DCIA) or its branches are also reported [11,24]. Bleeding can also result from the involvement of venous collaterals including periumbilical veins [25]. After IEA, DCIA is reported as a frequent injury site due to paracentesis [13,24]. As reported by Webster et al., some compelling evidence supports mechanisms other than vessel injuries such as the spontaneous rupture of venous collaterals, either esophagogastric or recanalized umbilical veins. The rupture occurs due to decompression of splanchnic circulations after paracentesis with a subsequent impaired reassessment of portal pressure and blood flow through the venous collateral [25].

Once the vessel lesions occur, the timing of clinical signs onset is variable, and hemorrhage can also manifest in the course of the procedure immediately after or delayed [14]. Given the objective formation of blood collection within the rectus sheath, injuries to the IEA have a rapid clinical onset compared to injuries of other abdominal wall vessels such as the DCIA and varices. Bleeding of vessels other than IEA may be more challenging to identify because of the lack of instantaneous hematoma and the only presence of hemoperitoneum as a clinical manifestation that can remain occult for up to 4 or more days [13,14,16,24,26]. In our cases, the hemorrhagic event occurred with a similar clinical presentation: after paracentesis was performed without complications, a rapid formation of a palpable, painful, firm, non-pulsatile abdominal mass occurred at the time of the needle extraction.

The diagnosis of vessel injury can be difficult, especially in delayed clinical manifestations. The vessel culprit can be investigated by ultrasonographic examination with eco-color-doppler to detect visible hematoma with or without active bleeding. The next step is a contrast-enhanced CT evaluation with optimal sensitivity and specificity to identify the bleeding site [8]. Angiogram evaluation under fluoroscopic guidance can also be used, especially when other imaging is negative [13,22]. The use of explorative laparotomy is described but exceptional [27].

In order to reduce the risks related to the procedure, an appropriate technique is undoubtedly required in carrying out paracentesis, and the optimization of risk rates is possible when the procedure is driven by an experienced operator under ultrasonographic guidance [28,29]. These findings also suggest that careful evaluation of the amount of fluid to drain is crucial for patients undergoing repeated LVP over time, especially in the presence of renal impairment and advanced liver dysfunction. Therefore, a close periprocedural follow-up is required, although this is never possible in outpatients discharged soon after the procedure unless a complication takes place [24].

Identifying the correct anatomical landmarks to perform the procedure is essential, both through traditional semiotics and instrumental ultrasound assistance. To avoid vessel injury, the needle should be inserted within a safe area, which is the left lower quadrant at the third lateral of the distance between the midline and anterior superior iliac spine, away from visible superficial venous vessels [20]. In a letter, Siau discussed how anatomical landmarks for performing paracentesis were still a controversial topic, especially in the case of a non-ultrasound-guided procedure. Although ultrasound guidance is always preferable, it is advisable to consider an ideal anatomical landmark to minimize risks without instrumental support. The authors describe the optimal point of needle insertion as the contralateral McBurney's point, which is at the third distal of the line between the anterior superior iliac spine and the umbilicus on the left side of the abdomen [30]. Following this recommendation, in our unit, paracentesis is performed by identifying the contralateral McBurney point and under ultrasonographic guidance to detect critical anatomical structures and avoid erroneous needle insertion.

Mercaldi et al. conducted an observational prospective cohort study using data from a hospital database to estimate the role of ultrasound guidance on the risk of bleeding after therapeutic paracentesis. Based on this analysis, ultrasound guidance was associated with a 68% reduction in the risk of bleeding complications from paracentesis and lower rates in the costs and length of hospitalization were observed. Furthermore, these findings demonstrate a higher benefit from ultrasound guidance in a cohort of outpatients [21].

Some authors have suggested the use of doppler empowered ultrasonography to localized IEA [18,20]. Doppler techniques can be a further aid in identifying the needle insertion site. However, it should be observed that minor vessels may not be visible at an ultrasonographic examination. Often the iatrogenic injury involves one of these branches rather than the parental vessel [25].

Therefore, even a procedure performed according to the reference standards can evolve as complicated. As discussed above, the risk of complications increases as the liver disease worsens. Although a complication during a therapeutic act is statistically expected in a certain percentage of cases, managing it can be challenging, and choosing the correct treatment to address the situation can be difficult.

Abdominal wall hematoma treatment options include a conservative approach, transcatheter embolization, percutaneous thrombin injection (preferred for pseudoaneurysm), and surgical ligation of vessels [13,14,22]. Transcatheter embolization is a highly effective treatment for IEA injuries (90%), and it should be the preferred option when available and when conservative treatment has failed [8,13,14,23,31]. Embolization seems superior to the surgical approach, burdened by not neglecting the risk of complications and mortality in cirrhotic patients [5,8,22]. Advanced liver disease is not only a bleeding complication risk factor, but also a potential obstacle in the management of the complication itself. For our patients, choosing the optimal treatment required a multidisciplinary evaluation that considered the risk–benefit balance of both the conservative and interventional strategies. As described in some reports, not all centers refer to a radiologic department with specific protocols for treating vascular lesions. Hence, the expertise of the operator and the hospital equipment and availability of materials and instrumentation is crucial [23,32,33]. If conservative treatment is preferred, it must be considered that the patient is exposed to further complications. In our case, the patient not undergoing an interventional procedure suffered a hemorrhage from esophageal varices and acute kidney damage. In a case report, Mehmood described a post-paracentesis intra-abdominal hematoma that occurred in a cirrhotic patient and was treated conservatively. Subsequently, the patient developed an obstruction of the small intestine due to the mass effect of the hematoma [34]. In our experience, both conservative and interventional approaches have been considered.

The hospitalization length was significantly different: the embolized patient was discharged after five days; the second patient was discharged after twenty-seven days because of variceal bleeding requiring pre-emptive TIPS placement.

Both our patients underwent TIPS placement.

Although still a controversial finding, the most recent studies seem to associate TIPS with a lower mortality and complication rate at 12 months compared to repeated LVP [4]. Therefore, identifying the correct timing for the TIPS placement can be crucial to reducing the risks of repeated LVP in refractory ascites.

Our report does not represent an innovative contribution since post-paracentesis abdominal wall hematoma are well-known and described bleeding complications. As a matter of fact, even a procedure, often considered "simple", can reveal troubles in a "difficult" patient, perhaps due to conventional expectations that things must always go for the best. Still, sometimes they do not go as they should. Accordingly, we aimed to describe two examples of real clinical practice relating to a rare but possible post-paracentesis adverse event, introducing the possibility of sharing and discussing medical cases, even of their controversial, complicated, or erroneous aspects: this could provide support to physicians and could help to identify any critical issues in job planning to improve performance and reduce the risk of further complications.

## 5. Conclusions

Although our reports do not represent an innovative contribution, we aimed to describe a different approach to hemorrhage after paracentesis. A gastroenterologist should remember that the hazard of complications arises with the progression of liver disease and in the presence of concomitant kidney disease. A well-performed procedure with correct landmark individuation and ultrasound-guided can reduce the overall rates of hemorrhagic complications. However, minor vessels may not be visible at ultrasonography, and often the iatrogenic injury involves one of these branches rather than the parental vessel. We would like to highlight the importance of prompt intervention in the case of hemorrhage, which is essential to avoid a poor outcome. A multidisciplinary approach is appropriate to face up this complication. Detailed information, monitoring, and supportive interventions are also necessary to ensure the most effective and safe treatments.

**Author Contributions:** All authors contributed equally. All authors have read and agreed to the published version of the manuscript.

**Funding:** This research received no external funding.

**Institutional Review Board Statement:** Not available.

**Informed Consent Statement:** Written informed consent was obtained from the patients to publish this paper.

**Data Availability Statement:** Not available.

**Conflicts of Interest:** The authors declare no conflict of interest.

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
