# Peer review of "Lights and Shadows of Paracentesis: Is an Ultrasound Guided Approach Enough to Prevent Bleeding Complications?"

_livers, doi:10.3390/livers3010004_

Round 1

Reviewer 1 Report

Dear author/ 

thanks to you for considering Liver journal for your manuscript that is valuable to readers . It need reformulation , organization . You should mention cause of bleeding concerning patients condition or the intervention. also cause of bacterial pneumonia of the second case. 

Author Response

RESPONSE TO REVIEWER 1

Point 1: Dear author thanks to you for considering Liver journal for your manuscript that is valuable to readers. It need reformulation, organization. You should mention cause of bleeding concerning patients condition or the intervention. also cause of bacterial pneumonia of the second case. 

Thank you for the request. Both patients underwent post-paracentesis hemorrhagic complications. On page 3 (line 109-111), we better explained the pneumonia complication.

Reviewer 2 Report

This is a  good summary of the bleeding complications presented by Patturelli et al. Tables and images are well done. This is a well discussed topic and I am not sure the readers of this journal might find it novel.  I have a few suggestions for the authors

1. Format of the discussion is not appropriate. I would suggest a more traditional approach i.e. epidemiology, clinical features/etiology/management/future directions. In the current format the authors go back and forth between all these components making it a difficult read.

2. Would suggest authors discuss the role of TEG/ROTEM in risk stratification of bleeding.

3. I am not sure why TIPS is being discussed in the management of refractory ascites in an article focusing on bleeding in paracentesis. I would suggest that the authors reconsider this section.

Author Response

RESPONSE TO REVIEWER 2

This is a good summary of the bleeding complications presented by Patturelli et al. Tables and images are well done. This is a well discussed topic and I am not sure the readers of this journal might find it novel.  I have a few suggestions for the authors

Point 1: Format of the discussion is not appropriate. I would suggest a more traditional approach i.e. epidemiology, clinical features/etiology/management/future directions. In the current format the authors go back and forth between all these components making it a difficult read.

Thank you for your suggestion. We modified the format according to the provided indications. See discussion paragraph.

Point 2: Would suggest authors discuss the role of TEG/ROTEM in risk stratification of bleeding.

Thank you for your suggestion. We included on page 6 (line 205-220) a statement about the TEG/ROTEM methods.

Point 3: I am not sure why TIPS is being discussed in the management of refractory ascites in an article focusing on bleeding in paracentesis. I would suggest that the authors reconsider this section.

Thank you for your suggestion. We rearranged the paragraph according to your suggestions. See discussion paragraph.

Round 2

Reviewer 1 Report

Dear Author/

thanks for considering Liver journal. minor revision is required.

The abstract need rewriting to view the novelty of the idea The results and discussion need reformulation to make the reader catch the main idea.

with my best wishes

Author Response

RESPONSE TO ROUND 2 REVIEWER 1

Point 1: Dear Author thanks for considering Liver journal. minor revision is required. The abstract need rewriting to view the novelty of the idea The results and discussion need reformulation to make the reader catch the main idea.

Thanks you. We modified the abstract and rearranged the results and discussion according to your suggestions.
